# Racism and access to maternal health care among garo indigenous women in Bangladesh: A qualitative descriptive study

**Suban Kumar Chowdhury** *

Department of International Relations, University of Rajshahi, Rajshahi, Bangladesh

* skc_ruir@ru.ac.bd

**Data Availability Statement:** All relevant data are within the paper and its Supporting Information files.

**Funding:** The author received no specific funding for this research.

## Abstract

Racism as social determinant of health significantly affects Indigenous women's maternal healthcare access. This study uses Jones' 'Three Levels of Racism' theory and an intersectional lens to explore how racism shapes the experience of maternal health care access among Garo Indigenous women in Bangladesh. Semi-structured interviews were conducted with 24 women of diverse backgrounds and pregnancy statuses using snowball sampling. Thematic analysis, incorporating inductive and deductive approaches, was employed for data analysis. The findings reveal a significant deviation from Jones' theory regarding the level of internalized racism within the specific context of Garo Indigenous women's experiences. Jones' theory usually focuses on how racism is internalized due to institutional and personally-mediated factors. On the contrary, this study uncovers a unique theme: 'women agency.' This theme emerges as a robust response among the Garo Indigenous women to their encounters with institutional and personally-mediated racism, highlighting their cultural resistance and resilience. The findings suggest that the complex relationship between these two forms of racism contributes to the strengthening of agency among Garo Indigenous women. Their agency stems from avoiding hospitals that disrespect their culture, manifesting their cultural resistance practice against the encountered racism at the institutional and relational levels. To increase biomedical healthcare access among Garo Indigenous women, it is recommended to address racism through intercultural competency training with the 'cultural safety' 'cultural humility' approach. This approach would foster inclusivity and empowerment, recognizing the agency of Garo women in healthcare decisions. Additionally, it would facilitate constructive dialogues between clinicians and Garo Indigenous women, acknowledging the shared experiences of racism within the latter group.

## Introduction

Racism, defined as unfair treatment based on ethnic or cultural origin [1], significantly affects the maternal health outcomes of Indigenous women [2, 3]. Although extensively studied in prior research [2, 4, 5], the specific effects of racism on Indigenous women's access to maternal healthcare have received limited attention within Indigenous women's health studies [6].

**Competing interests:** The authors have declared that no competing interests exist.

Research from various countries indicates higher rates of maternal morbidity and mortality associated with racism [7–9]. Instances of racial discrimination against Indigenous women by healthcare providers are evident in both India, the United States, and Mexico [10–14]. Bio-medical maternity care often exhibits racial bias and cultural insensitivity, resulting in trauma, mistrust, and reduced motivation among Indigenous women to seek healthcare [4, 15]. The intersection of gender, economic class, and other identities with racism further complicates Indigenous women's healthcare experiences [3, 16, 17], highlighting racism as a significant social determinant influencing their access to maternal healthcare [2, 18]. Racism as a system of privilege and oppression operates across institutional, personally-mediated, and internalized levels, resulting in an unequal distribution of power [19]. This systemic racism perpetuates stereotypes, bias, and unequal access to resources [19–21], highlighting the absence of cultural safety [22, 23] and cultural humility [24] approaches at the institutional and relational levels. Consequently, racism-driven discriminations have a substantial impact on maternal healthcare access [20, 25–29]. The patient-perceived discrimination scale is a valuable tool for investigating experiences within Indigenous communities [29]. Additionally, understanding racially biased provider attitudes, particularly implicit bias, enhances awareness of interpersonal discrimination [30–32].

Racism against Indigenous identity is gaining recognition [33] but remains overlooked as a health issue in South Asian countries, including Bangladesh [34]. Research reveals that racism significantly impacts Indigenous South Asian women's maternity care access, leading to health disparities and inadequate support [35–37]. In Bangladesh, limited research focuses on Indigenous women's experiences with racism and maternity care [38], in contrast to global research in countries like the United States, Mexico, and India [7, 32]. Most research on healthcare access and racism centers on patients' perceptions of provider bias or implicit discrimination [39–44], with few exploring its direct health effects [39, 45]. These studies often lack a context-specific understanding of intersectionality and overlook the perspectives of Indigenous women who face compounded prejudice and implicit bias due to various aspects of their identity such as gender and economic class. To comprehensively grasp Indigenous women's maternity care experiences, it is imperative to consider their diverse identities within distinct social contexts [41].

This study examines how racism impacts Garo Indigenous women's experience of accessing biomedical maternal healthcare in Bangladesh. The Garo community is one of the largest Indigenous communities in Bangladesh [46], and one of the few close-knit communities worldwide [47]. They follow a unique 'matrilineal-matrilocal' family structure [47]. In this arrangement, family lineage and property are traced through the mother's line [48]. Newly married couples reside with or near the wife's family, which grants women higher status and authority within the household and community [49]. Since 2011, however, the government's postcolonial policies labeled the Garo community as 'tribal' instead of 'Indigenous' based on their culture and language, resulting in hierarchical distinctions, restricted access to rights and resources, and the reinforcement of cultural inferiority and colonial exclusionary practices [50–57]. Despite their limited access to biomedical maternity care, racial biases in healthcare remain understudied [58–60]. Additionally, the scarcity of disaggregated data on indigenous women's healthcare access [36] underscores the necessity of this study.

The study employs Jones' 'Three Levels of Racism' framework [20] along with Crenshaw's intersectional lens [17] to frame and analyze the themes emerged from Participants' discussion. Jones' model categorizes racism into three levels: institutional, personally-mediated, and internalized [20]. This breakdown helps providing a deeper understanding of the complex impact of racism on maternal healthcare access. Unlike models that primarily focus on the social construction of race or ethnicity and the structural factors perpetuating racial

inequalities [61–63], Jones' comprehensive approach offers a holistic understanding. It suggests that institutional and personally-mediated racism contribute to the internalization of racism within the context of maternal healthcare access [20, 29]. Given its strong theoretical foundation, Jones' model is particularly relevant for our study.

Studies such as Janevic's research on Romani women [45] and similar work with Indigenous women [32, 64–66], provide significant insights into the influence of racism on maternal healthcare access. However, these studies often neglect a crucial aspect: the intersection of racism with various identities, including gender and economic class. Despite its importance, the impact of this intersectionality on maternal healthcare access is frequently underemphasized [17, 67]. Maternal health research [68, 69] underscores the significance of intersectionality, as it explores how social identities intersect to shape experiences, exacerbate healthcare disparities, and reveal interconnected systems of oppression [17]. Hence, in alignment with Jones' 'Three Levels of Racism' theory [20], the current study utilized intersectionality [17] as an analytical framework. It facilitates a clear understanding of how racism intersects with gender and economic class, leading to diverse experiences of discrimination that influence maternal healthcare access [17, 67, 70]. This approach enhances the comprehension of healthcare inequalities and provides valuable insights for addressing the impact of racism on Garo Indigenous women's access to maternal healthcare.

## Materials and methods

### Settings

The study was conducted in Kalmakanda, a sub-district in northern Bangladesh. It primarily focused on the Garo people living in the unions of Lengura and Nazirpur (See Fig 1 for study

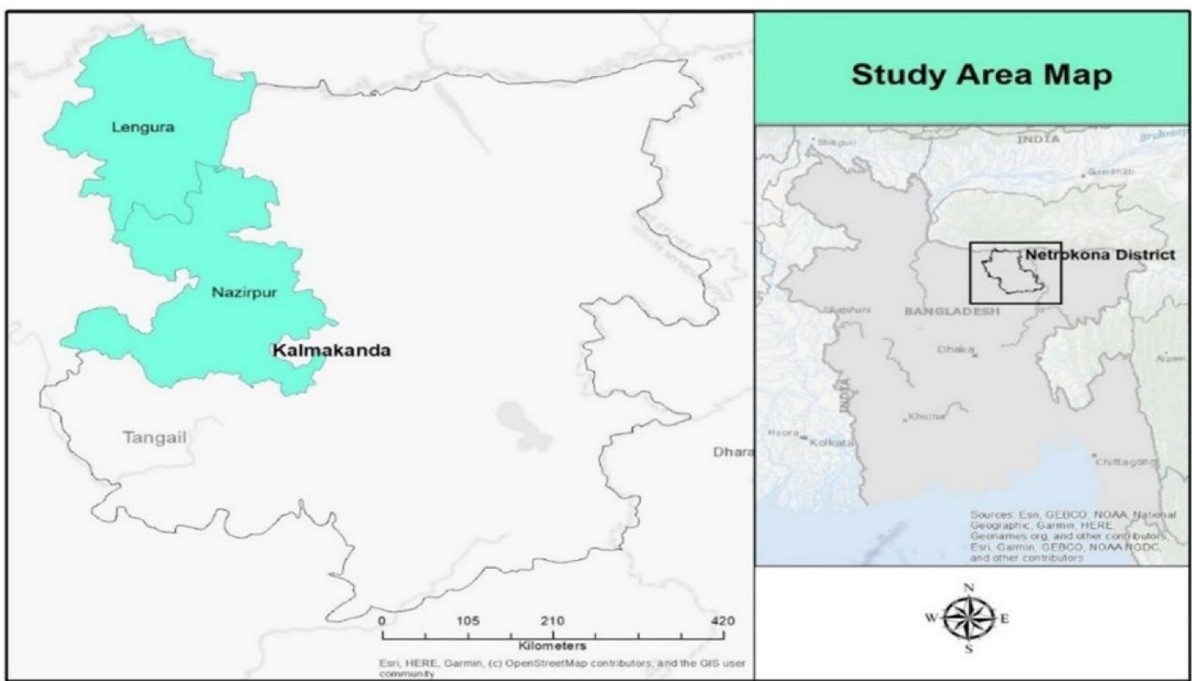

**Fig 1. Study area map.**

area map) [71]. Garo Indigenous women in these unions faced challenges as they had limited access to maternal healthcare facilities, lacking Union sub-centers and community clinics [72, 73]. While the Upazila hospital had medical staff, including non-Garo male gynecologists, it is noteworthy that few Garo women sought care there. This situation prompted questions about the potential influence of racism in this context.

The author of the study was a male and held the role of a researcher and primary investigator. He was from the mainstream non-Garo community in Bangladesh, living outside of the study area. This made him a stranger when it came to the place where the research was done and the Participants who were part of it. The research site involved interactions with Garo Indigenous women seeking maternal healthcare. Being an outsider to the Garo community and coming from a different culture, the author emphasized the significance of culturally relevant approaches when engaging with the interviewees- Garo Indigenous women. To achieve this, the author hired female Garo research assistants on a paid basis to collect data since they had a cultural affinity with the Participants, helping to establish cultural understanding and trust. This decision was based on the knowledge that cultural insiders could better connect with the Participants and conduct the research responsibly [74]. The author conducted three training sessions covering roles, rapport-building, and ethical considerations. Practical training included two demo interviews, enabling research assistants to gain the necessary skills for conducting interviews aligned with research objectives.

## Study procedures

This research was guided by the qualitative descriptive approach [75]. The qualitative descriptive approach is a widely used methodological approach, especially within practice disciplines [75]. This approach was selected for this study as the technique of inquiry because it was typically applied to research driven by exploratory questions [76]. The lack of in-depth research on maternal healthcare access among Garo Indigenous women led to a need for an in-depth exploratory study. The qualitative descriptive approach is notable for employing everyday language to create detailed summaries of findings [75]. It facilitated detailed summarization of experiences or phenomena [75], assisting in knowledge development in emerging fields [77]. Given the nascent state of research on Indigenous women's maternal healthcare access in Bangladesh [37, 59], the outcomes of this qualitative descriptive approach formed a crucial basis for future research aimed at a broader theoretical understanding of the experience of racism among Indigenous women accessing maternal healthcare in Bangladesh.

## Sampling and recruitment

The study used intersectionality-informed methods [78] to ensure a diverse sample by considering Participants' overlapping identities and experiences. This approach captured the complexities of their lives and how social factors shaped their perspectives [78]. Thus, beyond traditional categories, the selection process considered identities as multidimensional, leading to a comprehensive understanding of Participants' perspectives [17]. Purposive and snowball sampling techniques, as the intersectionality-informed methods [79], were used to enroll 24 Garo Indigenous women (n = 12 from Lengura Union and n = 12 from Nazirpur Union) [80, 81]. It helped to address the challenges in identifying pregnant and new mothers [80]. By combining purposive and snowball sampling methods and considering factors such as age, literacy, economic class, and pregnancy conditions, the study ensured diversity. It also helped to represent the diverse perspectives of the Participants. The author recognized historical challenges faced by Indigenous members in proving their Indigenous membership and did not require enrollment proof [82].

## Development of the Interview Guide

The author collaborated with Shila Gagra, a Garo school teacher in the study area during the study. Shila voluntarily provided valuable input in developing interview questions, ensuring cultural relevance. She also assisted in recruiting four Garo female research assistants and disseminating study findings.

## Data collection procedure

Data were collected from February to March 2021 through semi-structured interviews and open-ended questionnaires, exploring maternal healthcare experiences and practices. Interviews lasted at least 50 minutes, were audio-recorded, and field notes were taken. The comprehensive training of the research assistants and their efforts in the structured data collection process ensured the acquisition of detailed and valuable insights into the respondents' experiences with maternal healthcare access. To ensure rigorous data collection, research assistants followed strategies of encouraging Participants for open expression [83]. They performed member-checking with willing Participants, offering to provide a summary of findings for their review and feedback. This strengthened trust and credibility between the research assistants and Participants.

## Data analysis

Braun and Clarke's six-step framework was used to guide the thematic analysis in this research, incorporating both inductive and deductive approaches [84]. The rationale for employing both inductive and deductive methods lies in their complementary nature, as they allowed for the use of theory to understand the data while also permitting the data to generate key themes/concepts that speak to the theory. Integration of both methods allowed an evaluation of Jones' 'Three Levels of Racism' theory in the context of Garo Indigenous women, leading to novel insights that contribute to existing knowledge.

   To enhance both the reliability and coherence of the findings, collaborative efforts were made with research assistants throughout the data analysis process. The interviews underwent transcription in Bengali, followed by translation into English, and cross-checking by both the author and research assistants. Immersion within the transcripts aimed to achieve a comprehensive understanding of racial experiences in the context of Jones' 'Three Levels of Racism' theory, with an openness to emerging themes. An inductive approach was employed to generate initial codes within each transcript, forming the basis for a systematic search for themes guided by Jones' theory, validated through collaboration with research assistants. These themes were defined, named, and aligned with Jones' theory, recognizing the emergence of new themes. The themes were presented to the participating Garo Indigenous women to ensure their alignment with the Participants' perspective and viewpoint. Finally, the author wrote the results section using the identified themes and relevant verbatim quotes from the transcripts. A summary of the results, written in Mandi language through collaborative efforts with research assistants, was shared with Participants to gather their feedback for the final draft. Once consent was obtained, the author revisited the study area, invited all Participants, and presented the final version of the research findings. Shila Gagra and research assistants assisted the author in this process.

## Trustworthiness

To enhance the reliability of this study, the author verified the accuracy of codes and themes through collaborative discussions with the research assistants. Additionally, for credibility, the

codes, themes, and research results were reviewed by study participants who provided valuable feedback. These measures underline the author's dedication to producing trustworthy research findings, with an emphasis on both reliability and credibility in the study's outcomes.

### Ethical consideration

This study, conducted as part of the author's MSc thesis at the Asian Institute of Technology (AIT), Thailand, received ethics approval from the AIT Ethics Committee. There was no local ethics committee from which the author required to seek approval. Before conducting interviews, Participants and their husbands, as the local guardian, were informed about the research and provided verbal consent, following the approved approach. Research assistants from the Garo community established rapport and addressed concerns directly, leading to higher-quality approval. The prioritization of oral consent enhanced participant comprehension, engagement, and well-being, as reported by the research assistants, ultimately increasing the validity and reliability of the study. Confidentiality and voluntary participation were emphasized throughout the data collection process, ensuring Participants were free to withdraw without consequences. Pseudonyms were utilized to protect anonymity.

## Results

### Participants' characteristics

In-depth interviews (IDIs) were conducted with twenty-four Garo women (Please see Tables 1 and 2 for Participnats' profile). They typically expressed dissatisfaction with hospital-level

**Table 1. Participant profiles with Pseudonyms from Lengura Union.**

| Participant's pesudonyms | Age | Household economic status | Educational level | Pregnancy diversity |
|---|---|---|---|---|
| L1 | 21 | Low-class | Illiterate | She had 2 children and was in the third trimester of her pregnancy during the interview. |
| L2 | 30 | Middle-class | Illiterate | She had 4 children and her most recent childbirth occured within eight months before the interview date. |
| L3 | 22 | Middle-class | Illiterate | She had 3 children and was in the first trimester of her pregnancy during the interview. |
| L4 | 25 | High-class | She finished primary school education | She had 1 child and was in the first trimester of her pregnancy during the interview. |
| L5 | 19 | Middle-class | Illiterate | She was in the second trimester of the pregnancy during the interview and had no previous experience of pregnancy. |
| L6 | 20 | Low-class | Illiterate | She had 1 child and was in the second trimester of her pregnancy during the interview. |
| L7 | 27 | Low-class | Illiterate | She had 3 children and her most recent childbirth occured within five months before the interview date. |
| L8 | 20 | Middle-class | Illliterate | She had 2 children and was in the second trimester of her pregnancy during the interview. |
| L9 | 21 | High-class | She finished primary school education | She had no previous experience of pregnancy and her most recent childbirth occured within 2 months before the interview date. |
| L10 | 23 | Low-class | Illiterate | She had 2 children and her most recent childbirth occured within nine months before the interview date. |
| L11 | 19 | High-class | She finished primary school education | She had no previous experience of pregnancy and was in the second trimester of her pregnancy during the interview. |
| L12 | 26 | Low-class | Illiterate | She had 3 children and was in the third trimester of her pregnancy during the interview. |

**Table 2. Participant profiles with Pseudonyms from Nazirpur Union.**

| Participant's pesudonyms | Age | Household economic status | Educational level | Pregnancy diversity |
|---|---|---|---|---|
| N1 | 20 | Middle-class | Illiterate | She had 1 children and was in the second trimester of her pregnancy during the interview. |
| N2 | 20 | Middle-class | Illiterate | She had no previous experience of pregnancy and her most recent childbirth occured within four months before the interview date. |
| N3 | 22 | High-class | She finished primary school education | She had 1 child and was in the third trimester of her pregnancy during the interview. |
| N4 | 20 | Middle-class | Illiterate | She had no previous experience of pregnancy and her most recent childbirth occured within three months before the interview date. |
| N5 | 24 | Low-class | Illiterate | She had 2 children and her most recent childbirth occured within six months before the interview date. |
| N6 | 20 | Low-class | Illiterate | She had no previous experience of pregnancy and was in the third trimester of her pregnancy during the interview. |
| N7 | 21 | Low-class | Illiterate | She had 1 child and was in the second trimester of her pregnancy during the interview. |
| N8 | 23 | Low-class | Illiterate | She had 2 children and was in the second trimester of her pregnancy during the interview. |
| N9 | 20 | Low-class | Illiterate | She had 1 child and her most recent childbirth occured within seven months before the interview date. |
| N10 | 21 | Low-class | Illiterate | She had no previous experience of pregnancy and her most recent childbirth occured within three months before the interview date. |
| N11 | 20 | Middle-class | She finished primary school education | She had 1 child and her most recent childbirth occured within six months before the interview date. |
| N12 | 24 | High-class | She finished primary school education | She had 1 child and was in the third trimester of her pregnancy during the interview. |

healthcare access. Thematic saturation achieved after nineteen IDIs, signifying the point where no new themes or insights emerged. To deepen understanding, an additional five IDIs were conducted. The Participants actively engaged and interacted effectively with the research assistants. Their ages ranged from 19 to 30 years, and they had diverse educational backgrounds and income levels. Most of them had at least one child. The author categorized their household income into three classes: high, middle, and low, based on their monthly household income in comparison to the national average of monthly household income, which was 15,988 BDT as per Bangladesh Bureau of Statistics (BBS) data [85].

Ten themes were identified through open-coding, categorized into three main groups and analyzed using an intersectional approach (Fig 2). These three categories are: 'institutional racism,' 'personally mediated racism,' and 'women's agency.'

The first two categories correspond to Jones' institutionalized and personally mediated racism levels, which serve as frameworks for understanding the experiences of Garo women. These women consistently mentioned feeling disrespected and unfairly treated by hospital staff, highlighting their sense of mistreatment. While this concern was common among the participants, the author acknowledges the presence of individual nuances. This underscores the importance of addressing and enhancing the overall maternity care experiences of Garo Indigenous women.

The third category, 'women's agency,' emerged as a response to the first two categories. Jones' theory suggests that institutional and personally mediated racism leads to the internalization of racism [20, 45]. On the contrary, in this study evidence demonstrates that these two levels of racism strengthen Garo Indigenous women's agency of cultural resistance practices against the encountered institutional and personally mediated racism.

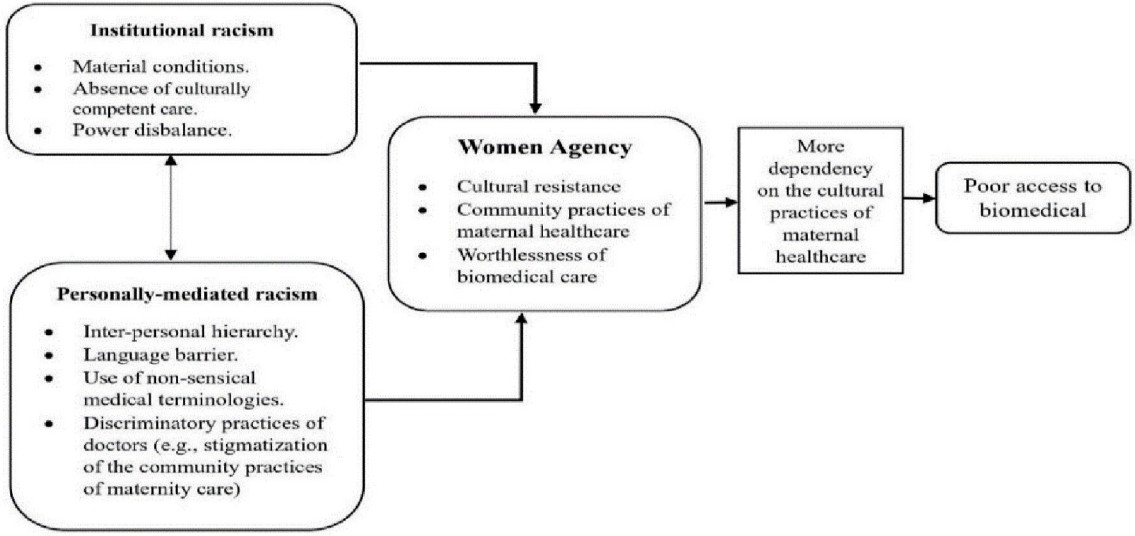

**Fig 2. Institutional racism, personally mediated racism, and women agency in maternal healthcare.**

### Institutional racism

Institutional racism is evident in material conditions, culturally inappropriate care, and resulting power imbalances. Regardless of economic class and education level, Participants faced challenges accessing maternal healthcare due to clinicians' claims of insufficient resources and power imbalances. For example, one participant reported clinicians' using the excuse of unavailability of medical facilities and limited supplies for discriminatory care at the hospital. As she typically discussed:

> During my last pregnancy, I saw a big difference in how doctors treated non-Garo women compared to me at the hospital. They got good care with all they needed, but the doctor said they didn't have enough for me. I was surprised when a non-Garo woman got proper care for the same problem. In our community, it's hard to access good medical facilities, so getting the care we need during pregnancy is tough. Doctors often use shortages of machines and staffs as excuses, making things even harder for us (N4).

This Participant described discrimination in care compared to non-Garo women at the hospital, suggesting healthcare service disparities based on Indigenous identity. Non-Garo women received proper care, while the Participants faced excuses about insufficient resources, raising concerns about institutional racism. Reflecting similar views with other Participants, another participant felt ignored by clinicians when seeking biomedical care due to their focus on non-Garo patients. As she reported: "due to the gynecologists' preference for non-Garo women, pregnancy care from hospital becomes impossible for us. But we don't expect this (L8)". When asked if experiencing discrimination at the hospital prevented Garo Indigenous women from seeking maternal healthcare, another woman typically responded: "Yes. . . Honestly, I can't recall a time when I went to a doctor's office for maternal care after facing discrimination during my first visit (L3)" As this Participant highlights, lacking access to equitable healthcare treatment often translates to limited access to medical professionals. Consequently, the women interviewed commonly pointed out inadequate services, discriminatory treatment, and clinicians' preference for local women.

Furthermore, all the Participants consistently voiced concerns regarding the hospital's lack of culturally appropriate care. This was particularly troubling for them as receiving biomedical care conflicting with their community's maternal healthcare practices was not acceptable. As one of them typically shared:

> I felt deeply troubled by the lack of alignment between the available healthcare at the hospital and our community's pregnancy healthcare practices. Following our cultural traditions is essential to us, and it was challenging for me to comprehend and accept healthcare practices that didn't respect our customs. It left me feeling confused and uneasy. (L6).

This participant's troubled, confused, and uneasy feelings indicate a dissonance between their expectations and the care provided. Another participant also mentioned that culturally inappropriate care includes not being able to meet the birth attendant before delivering in a hospital, which goes against their cultural norms and expectations:

> I don't understand why the doctors won't let us meet the midwife before giving birth. It's essential for us, and they don't seem to care about our cultural needs. They mistreat us, saying they can't do anything because of the hospital's rules (N6).

In addition, another woman with a common experience similar to other Participants expressed about the birthing dress code: "we can't go inside the delivery room without wearing the hospital's dress, given by the doctors. It's unfair and against our norms related to birth delivery because many women already wear their clothes during delivery (L1)".

While discussing the reasons behind the absence of culturally appropriate care, the Participants generally pointed out issues such as clinicians' inadequate cultural training, lack of community engagement, and limited knowledge about traditional healing practices. As one of the Participants generally discussed:

> I think the doctors don't understand our ways and beliefs. They don't involve our community in decisions. Our traditions are ignored, making us feel disconnected. They don't know much about our healing practices, so they prefer their own ways (L12).

This participant discussed her frustration and dissatisfaction with the healthcare system's lack of cultural understanding and involvement of their community in decision-making. She also expresses disconnection and alienation due to disregarding their traditions and healing practices. Interestingly, she and other low-income women also stressed the importance of female clinicians. In contrast, many other Participants linked the absence of Garo female doctors to culturally inappropriate care, highlighting the gender sensitivity of maternal health issues:

> I am deeply concerned about absence Garo female doctors. The non-Garo doctors does not have any care and respect for us and seeking care from non-Garo male doctor is against out our culture because we can't share our health problem to a male doctor (L5).

In addition, unlike the Participants from the high-class household income group, the Garo women from the middle-class and low-class household income statuses highlight the hidden costs of seeking medical treatment, such as transportation expenses and expensive hospital food. The difference between the claimed cost and hospital charges raises concerns about the financial strain on participants from middle-class and low-income households.

Furthermore, the lack of control over travel and treatment costs compounds the burden on these women. The participants' frustration with the hospital's dismissive attitude toward their concerns reflects a systemic issue of healthcare institutions not adequately addressing the needs and challenges faced by Indigenous communities. As one of the Participants shared:

> Distance cost isn't all. Hospitals want money. Despite their claim, they don't charge 5 Taka (Bangladeshi currency). They demand more. Traveling makes you hungry, but food is expensive. We must spend all we earn if we go. We don't control travel and treatment costs. They ignore our issue. They said 'you have to follow the hospital rules' whenever we told them our problem. What else can we do but avoid it? (L2).

This participant's discussion serves as an illustration of the systemic challenges encountered by Indigenous women in their pursuit of affordable and equitable healthcare. Through the lens of financial cost-driven burden, her narrative showcases how institutionalized racism can persistently obstruct access to healthcare, particularly for marginalized communities such as the Garo women

## Personally-mediated racism

Garo "Participants faced personally-mediated racism from clinicians, evident in the inter-personal hierarchy, use of non-sensical medical terms, and discriminatory practices like stigmatizing community maternity care practices. Language-related racism was also present, with clinicians viewing Garo Participants' use of the Mandi language as problematic, leading to miscommunication and discrimination against indigenous language speakers for not using mainstream language:

> They (the clinicians) don't understand what we're saying. We don't want to be treated by him because he's aggressive and doesn't know how. Our Garo women went there with health issues like jak jachak dalya (swelling of feet and hands), wakkalna ha'sika (vomiting), matha betha (dazziness), Hapani (shortness of breath), and bikma chikki sadika (abdominal pain), but the doctor didn't listen. They give us medicine even though they don't understand (N5).

This participant shared troubling experiences with clinicians lacking cultural understanding and empathy, expressing frustration with their aggressive and dismissive behavior, leaving her feeling unheard and misunderstood. Other Participants reported similar experiences, even those from high-class households with primary education. For instance, bilingual Garo women faced discrimination at the hospital due to difficulty pronouncing Bengali words. Using Bengali in the hospital was emotionally and practically harder than in daily life, and clinicians' unfamiliar medical terminology added to their challenges. As one of the Participants shared:

> Though I can't pronounce Bengali words well, I can explain my health issues to doctors. But, I'm hesitant to go again because the doctors laughed at me when I talked about my health issues a few months ago. The doctors replied with medical jargon. He laughed again when I asked him to explain these. I felt insulted (L7).

As this Participant discussed, she encounters challenges as an Indigenous woman seeking healthcare. Despite her efforts to communicate in Bengali, the clinicians' dismissive attitude

and use of medical jargon created a barrier to understanding. Their laughter made her feel demeaned and disrespected. This reflects a broader cultural issue where language barriers and lack of empathy hinder effective communication and care. Moreover, participants raised concerns about how clinicians wielded power and control over the bodies of Garo women. They described situations where clinicians used force, performed unconsented vaginal checkups, and prevented families from following their cultural rituals involving placenta. As, they unanimously shared, these actions significantly hindered the delivery of respectful maternal healthcare. One mother reported: "Our culture forbids delivering babies by cutting the belly. They touch our private organ during mid-pregnancy visits. Permission was never sought. They forcefully violate our culture (N2)."

The participant's quote indicates that clinicians exerted power to ignore the Garo women's cultural norms for maternity care and pressured them to adopt biomedical care as a symbol of modernity. The women's accounts also revealed that clinicians' perception of community maternal healthcare practices stigmatized them, creating obstacles in accessing the hospital:

> The doctors don't listen to our cultural needs, instead blaming us for our customs, such as not cutting the belly and reciting the delivery mantra. They said we are responsible for our health complications, especially birthing complications (N5).

The Participants generally shared experiences of clinicians' discriminatory practices, stigmatizing them for adhering to community maternity healthcare. Many feared hospitals visit due to concerns of forced C-sections or vaginal incisions, violating cultural norms and reproductive health. Clinicians' racism blamed severe maternal health complications on community practices, disregarding Indigenous culture and pushing for biomedical control. Thus, the clinician-Garo mother hierarchy is revealed:

> Whenever to visit the doctor, they stigmatize us for our belief in our community norms related to pregnancy health care. Disrespecting our community practices of not cutting the belly for delivery, they forced us to cut the belly. In fact, the doctor decides how to deliver, not we (N9).

### Women agency

The sub-themes that came up within this category consist of cultural resistance, community practices of maternal healthcare, and perceived worthlessness of biomedical care.

Concerning agency, while discussing the effect of the previous experience of being stigmatized, a mother reported her agency in deciding to avoid seeking biomedical care:

> Now, I do things my way when it comes to taking care of myself during pregnancy. I don't go to the hospital because they don't understand us. They make me feel small and don't listen to what I say. (N5).

This participant's discussion reflects empowerment through embracing traditional practices rooted in her cultural heritage. Exercising agency means she actively made choices based on her cultural beliefs, not just accepting the mainstream healthcare approach. She shows resilience in navigating a healthcare system that often fails to understand her unique needs. Her decision to avoid biomedical care is a response to the stigma and lack of respect she experienced in that setting. When asked, in addition to clinicians' stigmatization and discrimination,

what else contribute to strengthening the agency of making the decision to avoid seeking bio-medical care, she replied about her community bonding and support as another source for that:

> No, it's not just because of doctors' blaming and discrimination. Also, because our community women who are supportive to each other. Our elders know so much, and I trust them. All these gave me courage to decide to avoid going hospital (N5).

Interestingly, the interviewed women linked their perceived agency to avoid going hospital with the worthlessness of biomedical care compared to community practices of maternity care. Institutionalized racism and clinicians' personally-mediated racism, shown in power hierarchies and stigmatization, influenced their decision to rely more on community practices. One participant shared:

> Hospital-level care has no value for me because they disrespect us. When I need help with my health, I go to our traditional healers. They know our beliefs and take care of us with respect. I want to keep our traditions alive and stay connected to who we are (L7).

This participant's experience of institutionalized and personally-mediated racism shows how it becomes a source of culturally empowered agency to internalize the comparative bene-fits of community practices in maternal healthcare. For this participant, feeling culturally humiliated at the hospital was something she viewed as a reason to opt for seeking traditional healthcare. When asked what would have happened if she had not been able to get cured of possible maternal health complications, she responded:

> I knew from my neighbor women and mother, who had their previous pregnancy experi-ences that if I follow our cultural practices, then there will be no pregnancy health illness. So, why should I visit hospital to be humiliated and encounter discrimination? (L7).

When asked about the maternal healthcare practices within their community, along with this participant, another woman typically shared:

> We have several norms for a safe pregnancy, including avoiding houses where someone has died, not using mouth to fan fire smoke while cooking, eating locally sourced and culturally specific foods, spending time outdoors, participating in traditional activities, and seeking solace in natural environments. We also follow rituals and ceremonies related to pregnancy and rely on medicinal herbs and plants for managing pregnancy discomforts, promoting healthy labor, and aiding postpartum recovery. Additionally, we seek care from traditional healers and midwives within our own community (L8).

This Participant reported the importance and her dependency on the community practices of maternal healthcare as the manifestation of her resistance practices against the clinicians' discriminatory practices as a means to offset the clinician's intention of stigmatizing her. Simi-lar to other respondents, another participant felt that as she encountered discrimination at the hospital, she would not have been able to resist such racism against her without increasing her dependency on traditional healers whom she found far better:

> I was able to avoid getting stigmatized by the doctors when I started following our com-munity norms of pregnancy healthcare, I had good health care without making further

visit to the hospital. . . ..I went to traditional healer to all my prenatal visits. . .I had it good (N11).

## Discussions

This qualitative descriptive study investigates the impact of racism on Garo Indigenous women's maternal healthcare experiences, employing Jones' 'Three Levels of Racism' theory [20] and an intersectionality approach [17]. Nevertheless, the study's results, as outlined in the previous section, reveal a significant deviation from Jones' theory regarding the three levels of racism within the specific context of Garo Indigenous women's experiences. Jones' theory usually focuses on how racism is internalized due to institutional and personally-mediated factors. On the contrary, this study uncovers a unique theme: 'women agency.' This theme emerges as a robust response to institutional and personally-mediated racism within the Indigenous women studied, highlighting their cultural resistance and resilience. The findings indicate that the interplay between two forms of racism contributes to the strengthening of among the Garo women, ultimately shaping their preference for community-based maternal healthcare. This represents a departure from previous research conducted by Jenevic et al. [45] on Roma minority women in the Balkans. In their study, Jenevic et al. found that institutional and personally-mediated racism had different effects, leading to the internalization of racism among Roma women. This internalization was linked to the emergence of psychological factors, particularly a decline in self-esteem and self-efficacy.

Additionally, the current study uncovers the persistent and multifaceted discrimination faced by Garo Indigenous women in accessing maternity care, driven by institutional racism [20]. Participants generally highlight issues like lacking culturally competent care and power imbalances due to Indigenous identity-based racism at the institutional level. The exclusionary practices in healthcare, justified by resource constraints, reinforce institutional racism. This finding adds to the prior studies that examined how Indigenous identity-based racial discrimination is ignored by institutions providing biomedical care services [3, 21]. In contrast, this study diverges from previous research conducted in Ethiopia, where Indigenous women faced difficulties in accessing maternal healthcare primarily due to the actual scarcity of healthcare resources [42]. However, the study involving Garo Indigenous women in Bangladesh underscores that the main obstacle they encounter is institutional racism within healthcare facilities, rooted in their Indigenous identity. This variation underscores how local circumstances play a significant role in shaping discrimination across different regions [41].

The author's use of intersectionality [17] within Jones' framework of institutional racism [20] to analyze participant accounts suggests the connected and overlapping aspects of discrimination driven by these dimensions of institutional racism that specifically affect Participants' access to biomedical healthcare. Intersectionality demonstrates how these women's social background, Indigenous identity, and gender interact [17]. Race and socioeconomic position interact in the Upazila hospital's care and resource distribution, as the Participants highlighted. Gender and Indigenous identity intersect to discriminate against these women in healthcare as seen by the lack of Garo female doctors and the preference for non-Garo male doctors. Hidden costs and financial burden are experienced by Garo women from economically underprivileged communities, demonstrating racial and financial bias. As the Participants discussed, institutional racism results from the clinicians' dominating posture in forcing them to follow hospital policies. This position shows a power imbalance between men and women and is based on race. These findings are well supported by previous studies highlighting the intersectionality among race, gender, and socio-economic status of Indigenous women [3, 16, 17].

The stories told by the participants clearly show the long-standing biases and unfair treatment that Garo Indigenous women often experience within the healthcare system. Jones' theses of personally-mediated racism [20] which takes into account inter-personal hierarchy and asymmetric power relations based on 'othering' [54], fits issues like inter-personal hierarchy, language obstacles, and discriminatory practices of clinicians as encounters by the Garo women. The power relations between biomedical institutions and Indigenous women regarding maternity care have been the subject of prior studies conducted in several nations [28, 32, 43]. In this study, this pattern is also observable. As a result of Bangladesh's health policy, which perpetuates colonial stereotypes by classifying Indigenous people as a separate tribe, clinicians are unable to fully comprehend the imbalances in power that exist with Garo women [56, 57], It prevents clinicians from understanding historical and contemporary contexts and identifying unequal power relations with Garo women. Thus, Participants' discussion suggests that clinicians following service guidelines reproduce colonial prejudices and stereotypes in an institutionalized context. This implies that the different forms of racism experienced by Indigenous Garo women often interconnect and mutually reinforce each other, which is relevant to the concept of cultural safety in medical practice [22]. Cultural safety in healthcare emphasizes the importance of creating an environment where individuals, regardless of their cultural background, can receive care that respects their identity, needs, and experiences [22]. The interconnection of these forms of racism highlights the need for culturally safe healthcare practices that actively address and eliminate discrimination against Indigenous women.

Furthermore, the clinicians' racist approach and lack of consideration for cultural norms during pregnancy and childbirth underscore the impact of race and culture on healthcare access. Other studies have shown how segregationist colonialism and assimilationist policies have influenced relations between public institutions and Indigenous communities [33, 44], creating a racialized cultural logic of intercultural connections [32]. This study adds to these findings, demonstrating that racism intersects with other forms of discrimination [17]. For instance, the dominance of Western medical perspectives and stigmatization of Garo women for following their community's maternity healthcare practices highlight the intersection of race, cultural norms, and colonial legacies, leading to gendered power hierarchies and clinicians' attitudes toward Garo women. Furthermore, the fear of coerced medical procedures and clinicians' decisions underscores how race, gender, and power dynamics intersect, reinforcing the vulnerability of Indigenous women in healthcare. These findings extend previous research on racial-ethnic discrimination's impact on women's health [29], deepening the comprehension of the discrimination Indigenous women encounter. In addition, such findings of the current research also align with a study on Indigenous women in southern Mexico, which also noted concerns about unnecessary medical procedures [14]. These shared findings highlight the widespread nature of these issues and the pressing need for healthcare system improvements.

Besides, the participants' narratives about 'agency' highlights its significant influence on their healthcare choices. Themes like cultural resistance, community practices, and perceived worthlessness of biomedical care show how it shapes their decisions and cultural preservation. They resist biomedical care due to stigmatization, embracing community-based practices to strengthen their cultural identity. This contrasts with a study from Mexico where women abandoned Indigenous practices for biomedical care [32]. The intersectional analysis highlights that the Participants' agency is not solely derived due to individual attitudes but also institutional practices, policies, and personally-mediated racism.

Therefore, the study suggests implementing an Intercultural Competency Training Program (CICTP) for clinicians with a focus on 'cultural safety' and 'cultural humility.' The 'cultural safety' approach in healthcare prioritizes culturally sensitive care, crucial for Indigenous

and marginalized communities [23]. Similarly, 'cultural humility' acknowledges intersecting identities and power imbalances at both institutional and personal and levels of racism [24]. Integrating these principles can fosters an inclusive and empowering environment for Garo Indigenous women, acknowledging their agency in healthcare decisions. Furthermore, it would help initiate constructive dialogues with between the clinicians and the Garo Indigenous women to address the latter group's shared experience of exploitation.

## Strength and limitation of the study

A significant strength of this study is its application of an intersectional approach, focusing on the perspectives of Garo Indigenous women to grasp their challenges in accessing maternal healthcare. It highlights their resilience in the face of racism and recommends cultural competency training for clinicians, advancing health equity for Indigenous women. However, its limitations include restricted generalizability, a small participant size, and a qualitative, narrative-based methodology that limits applicability to other populations.

## Conclusion

Racism is a vital maternity health concern for Garo Indigenous women, with institutional racism and personally-mediated racism manifesting in discrimination and stigmatization due to language and community cultural norms related to maternal health care. This study presents the multi-dimensional nature of racism, as discussed by the Participants, and how it impacts their maternal healthcare access. The findings emphasize the significance of intersectionality, showcasing how institutionalized racism and personally-mediated racism generates Garo Indigenous women's agency and thus, shape the experiences of maternal healthcare. Furthermore, it underscores the importance of recognizing and respecting intersecting identities of race, gender, cultural identity, and socioeconomic status to address power disbalances at both personal and structural levels of racism. It highlights the need for clinicians' intercultural competency training programs incorporating 'cultural safety' and 'cultural humility' approaches to create an inclusive and empowering healthcare environment for Garo Indigenous women that ensure equity in healthcare access.

## Supporting information

**S1 Checklist.**
(DOCX)

**S1 Dataset.**
(DOCX)

## Acknowledgments

The author would like to express gratitude to the anonymous reviewers for their valuable input. This article is based on the author's MSc thesis work, which was supervised by Dr. Joyee S. Chatterjee, Program Chair of Gender and Development Studies at AIT. The author extends heartfelt thanks to Dr. Chatterjee. Special recognition is given to Shila Gagra for her assistance in developing culturally relevant interview questions and recruiting four female research assistants: Manisha, Namita, Malita, and Tomita. The author sincerely appreciates their active supports throughout the research. Additionally, the author acknowledges Dr. Eisuke Saito, Dr. Anne Keary (Faculty at Monash University, Australia), and Dr. Melissa Barnes (La Trobe University, Australia) for their guidance on using inductive and deductive approaches in research.

## Author Contributions

**Conceptualization:** Suban Kumar Chowdhury.

**Formal analysis:** Suban Kumar Chowdhury.

**Methodology:** Suban Kumar Chowdhury.

**Supervision:** Suban Kumar Chowdhury.

**Writing – original draft:** Suban Kumar Chowdhury.

**Writing – review & editing:** Suban Kumar Chowdhury.

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
