## [Editor Report · Decision Letter 0]

14 Mar 2023

PONE-D-22-34953“They don’t even listen to our voices”: Ethnocentrism and Biomedical Maternity Care Access Among Garo Indigenous Women in BangladeshPLOS ONE

Dear Dr. Chowdhury,

Thank you for submitting your manuscript to PLOS ONE. After careful consideration, we feel that it has merit but does not fully meet PLOS ONE’s publication criteria as it currently stands. Therefore, we invite you to submit a revised version of the manuscript that addresses the points raised during the review process.

It has been very difficult to secure a peer review for your manuscript. Therefore, I decided to conduct an initial review and request major revisions.

The manuscript is unnecessarily long. Please, consider resubmitting a manuscript that is not more than 4,000 words. The results section has a lot of repeated information, even across different categories. 

You should write a shorter discussion section which addresses the following topics (do not use these as subheadings): a statement of the principal findings; strengths and weaknesses of the study; strengths and weaknesses in relation to other studies, discussing important differences in results; the meaning of the study: possible explanations and implications for clinicians and policymakers; and unanswered questions and future research.

Please, carefully review the instructions for authors before resubmitting your paper (https://journals.plos.org/plosone/s/submission-guidelines#loc-page-and-line-numbers).

At PLOS ONE, we recommend that authors use the COREQ checklist, or other relevant checklists listed by the Equator Network, such as the SRQR, to ensure complete reporting (http://journals.plos.org/plosone/s/submission-guidelines#loc-qualitative-research).

Abstract

Please, clarify if you used grounded theory or inductive thematic analysis. It looks to me that you used the latter.

You did not even mention the three levels of ethnocentrism in the results section of your abstract.

The conclusions in the abstract refer to two levels of ethnocentrism and mention a policy document that you did not mention before. Please, rewrite this section.

In academic English, the use of long sentences is highly discouraged. Long sentences reduce the readability of texts. Please, revise the English accordingly.

Background

You say your objective was to study disparities. However, you did not present any comparison with other groups. It looks like your study aimed to address the experience of Garo women in their access to maternal care. Please, revise.

Revise the paragraph with the definition of ethnocentrism. It is not clear.

The last paragraph is unnecessary. Please delete.

Methods

What is the 'intersectionality-informed method' you used? Please, explain it briefly when you mention it.

Table 1 belongs to the results section. In the results, you should start by describing the participants. Also, Tables 1 and 2 could be integrated into one table.

Please, better explain what are ‘Lengura and Nazirpur Unions of Kalmakanda Upazila.’ Remember, we have an international audience not familiar with Bangladesh.

You indicated Janevic's conceptual framework (methods) and then Anderson’s framework (in the Discussion) were used to create analytical categories. However, you described an inductive analysis. Please, be more concrete and specific in describing what you did.

Results

You mention that Figure 1 has five domains ‘for more transparent comprehension.’ However, in Figure 1, it seems like you used three categories to summarise results.

Overall, you present interesting quotes, but they have a lot of repetition. Please, consider making some quotes shorter to facilitate readability.

What does Ethnocultural mean? Perhaps use cultural.

What does Womanization mean? Please, consider not using this word if you could explain the idea with more accessible language.

Page 11, last paragraph. You said ‘Garo women's acceptance of the health practitioners' ethnic and occupational superiority may be the outcome of internalizing discriminatory attitudes encountered by the former group.’ This is an assumption. Please delete.

Last sentence on page 13. Interestingly, while showing the ethnocentric aspects of maternal care, the author reproduces these views, for example, when describing mothers and mothers-in-law as ignorant people who cannot provide proper advice to pregnant women. Is this assumption supported by data? It is unacceptable that you describe Garo mothers as ignorant and discuss the cultural competency of their care simultaneously. Furthermore, this topic should be part of the health literacy part.

Clarify the first sentence of Health System Factors. Also, it is unclear if you are addressing health system-level issues here because the content is mostly about institutional care.

Please, avoid repetition. For example, the information in the last paragraph of page 15 was already mentioned on page 9 as part of a different category.

The last paragraph of the results belongs to the discussion. Please delete.

Discussion

Why did you mention Anderson’s framework? This seems very unnecessary. Perhaps it is better to delete it.

How did you jump from three categories of ethnocentrism to five types of discrimination? Please, avoid this or explain the methods and present the results.

References 39-41 have nothing to do with the Upazila. They do not support your statement. 

We look forward to receiving your revised manuscript.

Kind regards,

Ivan Sarmiento

Academic Editor

PLOS ONE

Journal Requirements:

2. In the ethics statement in the Methods, you have specified that verbal consent was obtained. Please provide additional details regarding how this consent was documented and witnessed, and state whether this was approved by the IRB

 “NO”

7. Please include your tables as part of your main manuscript and remove the individual files. Please note that supplementary tables (should remain/ be uploaded) as separate "supporting information" files.

8. We note you have included a table to which you do not refer in the text of your manuscript. Please ensure that you refer to Tables 1 and 2 in your text; if accepted, production will need this reference to link the reader to the Table.

---

## [Author Response · Author response to Decision Letter 0]

24 Apr 2023

Dear respected reviewers,

Thank you very much for the valuable comments and guidelines. I tried my best to address all the review comments. I have attached a file 'Response to the Reviewers' listing all of the review comments and my responses to these. 

Thank you again for the kind supports.

Kind Regards.

The Author

---

## [Decision Letter · Decision Letter 1]

4 Jul 2023

PONE-D-22-34953R1“They don’t even listen to our voices”: Ethnocentrism and Biomedical Maternity Care Access Among Garo Indigenous Women in BangladeshPLOS ONE

Dear Dr. Chowdhury,

Thank you for submitting your manuscript to PLOS ONE. After careful consideration, we feel that it has merit but does not fully meet PLOS ONE’s publication criteria as it currently stands. Therefore, we invite you to submit a revised version of the manuscript that addresses the points raised during the review process.

We look forward to receiving your revised manuscript.

Kind regards,

Ivan Sarmiento

Academic Editor

PLOS ONE

Reviewers' comments:

Reviewer's Responses to Questions

**Comments to the Author**

1. If the authors have adequately addressed your comments raised in a previous round of review and you feel that this manuscript is now acceptable for publication, you may indicate that here to bypass the “Comments to the Author” section, enter your conflict of interest statement in the “Confidential to Editor” section, and submit your "Accept" recommendation.

Reviewer #1: (No Response)

Reviewer #2: All comments have been addressed

Reviewer #3: (No Response)

2. Is the manuscript technically sound, and do the data support the conclusions?

Reviewer #1: Partly

Reviewer #2: Partly

Reviewer #3: Partly

3. Has the statistical analysis been performed appropriately and rigorously? 

Reviewer #1: No

Reviewer #2: N/A

Reviewer #3: N/A

4. Have the authors made all data underlying the findings in their manuscript fully available?

Reviewer #1: (No Response)

Reviewer #2: Yes

Reviewer #3: No

5. Is the manuscript presented in an intelligible fashion and written in standard English?

Reviewer #1: No

Reviewer #2: No

Reviewer #3: No

6. Review Comments to the Author

Reviewer #1: The Title and topic can be changed a bit, the way is was presented now is more of tribal, ethnicity and ...

Read my comments carefully.

Reviewer #2: Dear author,

Thank you for the opportunity to review this paper. It is an important area of scholarly investigation.

The best academic writing is writing that is clear, confident, and adopting simple terms. Your paper feels complicated, heavy on jargon, and lacking flow. To address this please work with your supervisor to enhance clarity and readability.

In addition your paper requires a positionality section to situate you, in relation to the study.

The use of the term 'Indigenous' in your study context should always have a capital I. This is important in enhancing language inclusivity and anti-colonial approaches.

I ask why your three Garo research assistants are not named as fellow authors? There is a large movement internationally to recognize the cultural capital Indigenous researcher and liaison officers.

I am concerned there was no written informed consent with participants, only verbal. This needs to be explained.

I am also concerned the results of the study have not been taken back to the participants (or maybe the Garo research officers). Failure to share results is contrary to best practice principles of doing research with Indigenous communities and principles of Indigenous data sovereignty.

Finally your discussion seems simplistic, it will take more than one off training to improve maternity care. Exploring multi-level approaches and options such as Community controlled health services, Garo liaison officers, government commitment to improving health equity, other international policy drivers are all needed. I have provided some marked up comments but I believe this work needs significant attention prior to publication.

I wish you the very best.

Reviewer #3: The study addresses a central health problematic, crucial for indigenous preservation. The aim of analyzing Garo women health maternity care from an intersectional perspective is more than pertinent, and highlighting Garo women voices towards this problem is already part of the solution. Author can find below detail suggestions to achieve publications, as mayor issues are needed, related to clarity on the theoretical framework, the research design (qualitative approach / analytical approach), and results.

Introduction

Theoretical framework: mayor issues

1. Janevic’s conceptual model proposes three levels of racism, not ethnocentrism. This is a crucial difference, as ethnocentrism is itself a structural level of racism (Nelson y Prillentensky, 2010). Author should master the theoretical framework of the study, so it would be useful to revisit Janevic’s model, and even more Jhones`(2007) original piece, developed later by Janevic, in which he proposes those three levels for public health analysis.

2. the authors` option for intersectionality is pertinent for the problem. I suggest a clear differentiation of this concept from Janevic’s model, that does not address it (in abstract and introduction).

3. Cultural humility as well as cultural safety framework addresses intersectionality than the cultural competence approach. Cultural safety is an approach widely applied in the interpersonal (clinical), institutional (health centers) but also in a structural level (public health policies) that seems to be very relevant for the actual public policy context of Bangladesh summarized by the author.

Other minor theoretical issues:

4. The author affirms that Janevic´s model has been largely used to study maternal healtcare (p.5). Please cite those studies.

Context

1. The author contextualizes current situation of Garo people in Bangladesh. For the international community, it would be useful to clarify who and when where recognized as “tribal” and why it is a problem or constitutes not being recognized as indigenous population.

2. The text claims that “only 62% of Garo women receive skilled maternity care. ¿is the author referring to biomedical maternity care? In the context of the paper claims, it is important to recognize that community care and traditional medicine can be skilled maternity care as well, (Sarmiento et al, 2021).

Methodology

Major issues are related to the approach and analysis:

Approach: It is recommended to choose a qualitative approach in the methodology design to enhance coherence. This study seems proximal to a qualitative descriptive approach (Sandelowski, 2000), as it aims to describe a situation as it is lived and reported by the people experience it, with a low level of interpretation. This approach has a specific scope, and does not offer evidence required to give recommendations different than specific topics that needs to be more profoundly studied. It also guides the rest of the methodological design.

Analysis

Here, as well as in the abstract, clarify the analytical methodological approach, not only the theoretical approach. Did you do a thematical analysis? inductive or deductive? If so, include the references of the authors. (e.g. Braun, V. & Clarke, G. for Thematic analysis)

Other methodological minor issues:

Positionality and reflexivity : It is recommended to evidence positionality or reflexivity in the report. Consider to share the relation of the author with the research site/participants/research subject.

Participants

1. The intersectionality methods for participants selection requires clarification. Are participants selected based on diverse experiences of institutional and personal levels of racism experienced? Did de sampling intended diverse combination of age, literacy and pregnancy conditions or other particular socio-cultural identities?

2. I suggest to include in this section the selection criteria expressed in Table 1.

3. It would be Useful to present a detailed characterization of participants with pseudonyms or codes, in order to see their diverse intersectional features. This also allows to follow in the results the variation of their experiences.

Trustworthiness

Please report if you used particular strategies to enhance trustworthiness, such as triangulation, reflexivity, peer review, member checking or other.

Revise the clarity of this sentence: “The author noticed active engagement of all participants with good interaction with himself and the research assistants”. The author expressed earlier that only research assistances interviewed Garo women.

Results

This paper has a special value for sharing the experiences of Garo women whose health and dignity are put at risk. Their perspective needs to be highlighted in this section of the paper:

1. When results and discussion are presented in differentiated sections of a paper, results must not include a dialogue with other studies or the theoretical framework. That is reserved to the discussion section.

2. The author presents interpersonal level of racism –particularly clinicians` disdain towards Garo culture– as a cause for Garo women to assume biomedical as worthless, and therefore to dismissing biomedical healthcare. two points to revise on these findings.

a. Garo women unattendance to healthcare centers could be interpreted not as a lower agency indicator, but as a (cultural) resistance practice. If they were, as the author suggests, loosing agency, they would instead assist to healthcare centers and abandon their traditional practices. Community practices of maternity is a theme that could show empowerment and resistance of the Garo women. Therefore, it could be highlighted in the results as an independent theme, even more when author concludes in this regard.

b. The paper has the crucial aim of addressing racism. Please revise a possible implicit bias towards the biomedical model as it suggests that not having biomedical assistance is the same as not having skilled maternity care (p.4). Other studies have shown that traditional medicine, –e.g. being assisted by traditional indigenous midwifes in México (Sarmiento, 2021)– has no significant difference with biomedical assistance in health centers, in relation to health outcomes.

3. Consider to adjust themes names:

a. The theme “community practices of maternity” within “Personally mediated ethnocentrism” could be clearer if it is named accordingly to the discriminatory practices of doctors. (e.g. stigmatization of C.P of maternity)

b. instead of power dynamics (they are always present in social contexts), abuse of power or power disbalances. I assume this is particularly relevant in Garo culture and the place of women within the community? If so, this could be address in discussion.

4. Please clarify this result, as the fragment does not seem to be related to midwife meeting:

“Furthermore, they complained not meeting the midwife before hospital birth. She said

Doctors are not Garo women. No one is allowed to enter the delivery room without wearing the

hospital’s dress provided by them (the doctors). It's unfair because many women use these clothes,

already, during their delivery (N6, age 28).”

Discussion:

Limitations of the study do not belong to this section.

Adjust accordingly to theoretical precision. Consider to include the relevance of cultural humility or cultural safety within maternal healthcare in this context as these approaches address intersectionality and power imbalances in the personal and structural level of racism. The agency of the women who refuse to attend healthcare centers in current conditions, could be well receive in programs designed with this approach.

7. PLOS authors have the option to publish the peer review history of their article (what does this mean?). If published, this will include your full peer review and any attached files.

Reviewer #1: No

Reviewer #2: No

Reviewer #3: No

---

## [Author Response · Author response to Decision Letter 1]

22 Jul 2023

I attached a file title 'response to reviewers' comments'.

---

## [Decision Letter · Decision Letter 2]

15 Aug 2023

PONE-D-22-34953R2Racism and Access to Maternal Health care Among Garo Indigenous Women in Bangladesh: A Qualitative Descriptive StudyPLOS ONE

Dear Dr. Chowdhury,

Thank you for submitting your manuscript to PLOS ONE. After careful consideration, we feel that it has merit but does not fully meet PLOS ONE’s publication criteria as it currently stands. Therefore, we invite you to submit a revised version of the manuscript that addresses the points raised during the review process.

Please, in addition to addressing the reviewer's comments (attached), ensure that you have a proper review of the English of your manuscript.  Academic editors do not handle publication fees; please contact the journal office to discuss this with them. ==============================

We look forward to receiving your revised manuscript.

Kind regards,

Ivan Sarmiento

Academic Editor

PLOS ONE

Reviewers' comments:

Reviewer's Responses to Questions

**Comments to the Author**

1. If the authors have adequately addressed your comments raised in a previous round of review and you feel that this manuscript is now acceptable for publication, you may indicate that here to bypass the “Comments to the Author” section, enter your conflict of interest statement in the “Confidential to Editor” section, and submit your "Accept" recommendation.

Reviewer #3: (No Response)

2. Is the manuscript technically sound, and do the data support the conclusions?

Reviewer #3: Yes

3. Has the statistical analysis been performed appropriately and rigorously? 

Reviewer #3: N/A

4. Have the authors made all data underlying the findings in their manuscript fully available?

Reviewer #3: Yes

5. Is the manuscript presented in an intelligible fashion and written in standard English?

Reviewer #3: No

6. Review Comments to the Author

Reviewer #3: The author made significant effort improving the report of this pertinent study. There is still work to do, particularly in results and discussion. Considering the importance of sharing findings related to maternity healthcare scenario of Garo women as a risk for them and their culture, I encourage the author to consider the recommendations.

7. PLOS authors have the option to publish the peer review history of their article (what does this mean?). If published, this will include your full peer review and any attached files.

Reviewer #3: No

---

## [Author Response · Author response to Decision Letter 2]

6 Sep 2023

Dear esteemed reviewer,

I would like to express my sincere gratitude for your invaluable feedback on my manuscript. Your insights and recommendations have been immensely helpful in refining the quality and clarity of the research.

I am pleased to inform you that I have diligently addressed each of your suggestions, and the changes have been incorporated into the revised manuscript. To provide a clear overview of the modifications, I have compiled them, as below:

Reviewer's comments on Acknowledgement:

I strongly suggest you include at least in this section the acknowledgment of the Garo research assistants, whom not only participated in data collection, but also in the analysis and member checking strategy. Do the same with the teacher. 

RESPONSE: Dear respected reviewer, thank you for your thoughtful suggestion. In response to your recommendation, I have included acknowledgments for both the Garo research assistants and the teacher in the appropriate section of the manuscript. Their significant contributions to data collection, analysis, and member checking have been duly noted. Your feedback is greatly appreciated.

Reviewer's comments on Study procedure: 

• Review this phrase: “This study utilized a qualitative descriptive approach, commonly employed in semi-structured interviews”. Do you mean “qualitative descriptive approach, recollecting data with semi-structured interviews”?

• Author should expand the description of the method`s scope and utility, explained by Sandelowski: it aims to explore and describe a situation, generally understudied, as it is lived and reported by the people whom experience it. Remember that this approach has a specific scope. It is exploratory and it generally does not have evidence deep enough to support recommendations (as policies). Researchers discuss and conclude based on the data, However, they may point central phenomena to take into consideration in further studies or policy making.

• Sampling: review the sentence “ The participants were categorized into these three groups based on… “ ¿which groups?.

• Sampling. Table 1. 

This table in qualitative descriptive studies is important as it shows the variation of the sampling, highly valued in this kind of research. Variation in the participant´s group assures an expanded exploration of an understudied phenomena. Participant table could be helpful to indicate possible areas or populations that require further research. For this, we find this table in the results section. 

• Include in this section (sampling) only table 2, adding all the inclusion criteria.

• As author have made significant improvements in results and discussions highlighting intersectionality relevance, table 1 is more useful in the first part of the results. This table must be clarified as follows. 

-Please revise precise data of every participant, as in the current table we have mixed information.

-Briefly explain how Household economic status is determined in your context. 

-Clarify what “class five” means in your context ¿basic education or high school?

-Briefly explain what does “Self-educated” mean in Garo women context.

-Pregnancy diversity can be better understood if it is shared how many children have been born and if they are or not currently pregnant.

RESPONSE TO THE REVIEWER'S COMMENTS ON STUDY PROCEDURES:

1.Dear respected reviewer, thank you very much for your valuable comment. I revised the sentences as: This research was guided by the qualitative description approach. Qualitative description is widely used methodological approach, especially within practice disciplines.

2.Dear respected reviewer, I addressed this comment following your guidelines. [please see the ‘procedures’ section of the revised manuscript. 

3.Dear respected reviewer, thank you for your feedback. The sentence you mentioned has been revised as follows: "The author categorized their household income into three classes: high, middle, and low, based on their monthly household income in comparison to the national average of monthly household income, which was 15,988 BDT as per Bangladesh Bureau of Statistics (BBS) data."

I believe this revision provides a clearer explanation of how the participants were categorized based on their income levels.

[please see the result section, page 11] 

4.Dear respected reviewer, thank you very much for your valuable comment. I have addressed your valuable comments by making the following revisions:

 * I have relocated Table 1 to the first part of the results section, as suggested by the reviewer. This adjustment helps emphasize the variation in participant groups and the potential areas for further research, aligning with the goals of qualitative descriptive studies.

 * In response to your recommendation, I have included Table 2 in the sampling section, incorporating all the inclusion criteria. This ensures that the sampling section comprehensively presents participant characteristics.

 -I have revised and provided precise data for each participant, ensuring that the information in the table is clear and well-organized.

 -I have included a brief explanation of how household economic status is determined in the context of the study.

 -I clarified that "class five" in the context refers to primary level of education.

 -I have replaced the term "Self-educated" to illiterate as it will be easier to understand for the reader.

 -I have included information about the number of children each participant has and their pregnancy status to enhance the understanding of pregnancy diversity.

Reviewer's comments on ANALYSIS SECTION:

1.In this new version author reports doing content analysis. Even though this approach is highly used in qualitative descriptive analysis, results are presented in themes. There is a common confusion between content and thematic analysis in qualitative descriptive analysis and needs to be avoided. Vaismoradi, Turunen and Bonda clarify how to address this risk in their paper. Please adjust analysis, briefly highlining the process that makes it content analysis or thematic analysis, to address methodological coherence and transparency in the report. 

2.Also, clarify if it is a deductive or inductive process. Apparently, the author did a deductive analysis, based on Jones’ racism framework, combined with an inductive analysis, as there seem to have emergent categories (not addressed in Jones´ racism theory): Agency; Cultural resistance practice; Community practices of maternal healthcare (traditional medicine practices of maternal health?).

RESPONSE TO THE REVIEWER'S COMMENTS OF THE ANALYSIS SECTION:

1.Dear respected reviewer, I appreciate your valuable feedback, and I have made the necessary revisions to the analysis section to ensure methodological coherence and transparency. In the revised version, I have explicitly mentioned that I used 'thematic analysis' as the method for analysing the data. This adjustment helps clarify the approach taken in the study, aligning it with the methodological choice and addressing any potential confusion between content and thematic analysis. Thank you for highlighting this important point, and I hope the clarification improves the clarity of the report.

2.Dear respected reviewer, thank you for your insightful comments. I have addressed the issue of deductive and inductive analysis in the revised 'Analysis' section of the manuscript. I explicitly mention that I employed both inductive and deductive approaches in a complementary manner.

I clarified the rationale behind this choice, explaining that while I initially used a deductive approach based on Jones' racism framework, I also allowed for the emergence of additional categories that were not addressed in Jones' theory. This combination of deductive and inductive analysis enriches the understanding of the data and ensures that both the pre-defined and emergent themes are thoroughly explored.

I hope these clarifications provide a more comprehensive view of the analysis process in the study. Thank you for your valuable feedback, which has contributed to improving the rigor and transparency of the research.

Reviewer's comments on Trustworthiness:

consider include this section in the procedure as you develop multiple strategies to enhance rigor in your context: triangulation between assistants and researcher during analysis; member-checking of results with participants. Also include here positionality of author and the strategies used to enhance cultural sensitivity with the support of Garo women assistants and the community teacher.

RESPONSE

Dear respected reviewer,

Thank you for your suggestion regarding the inclusion of a 'Trustworthiness' section in the procedure. I appreciate your thoughtful input. I want to inform you that I have indeed included a separate section titled 'Trustworthiness' immediately after the 'Analysis' section in the revised manuscript. In this section, I have addressed multiple strategies employed to enhance the rigor of the study.

I also merged the section on 'Author's Positionality' with the section titled 'Research Site' to provide a more comprehensive understanding of the context and the strategies used to enhance cultural sensitivity with the support of Garo women assistants and the community teacher.

I believe these revisions have strengthened the manuscript, and I hope they align with your expectations regarding the presentation of trustworthiness measures. Your feedback has been instrumental in refining the study, and I am grateful for your guidance.

Reviewer's comments on the RESULT SECTION

Author improved significantly the results report. Fragments included about traditional medicine practices, and about women agency and trust in their culture are valuable. However, because of the way results are presented, there is a major theoretical incoherence that could have a simple solution: 

1.How does agency, Cultural resistance practice. and Community practices of maternal healthcare constitute internalized racism? It is exactly the opposite of internalized racism. If women had internalized racism (as in the Mexican study you cite on discussion), they would prefer to go the hospitals instead of their midwifes, they would not be proud of their culture, they would even believe they deserve bad treatment.

2.This emergent theme is crucial to understand and support Garo Culture and Garo women strategies to face racism at bio-medical centers. Protecting Garo Culture and women agency is as necessary as transforming the biomedical healthcare institution, from the theoretical framework reported. I strongly suggest to create an additional theme grouping these sub-themes, related to cultural strength or resistance towards racism.

3.Even though the author clearly has the intention to support Garo Women and Garo culture from racism, the model continues to suggest that more dependency on the cultural practices of maternal healthcare is a negative effect of institutional racism and internalized racism. therefore, it suggests cultural strength cause poor access to biomedical maternity care. In this regard consider that:

 -there is no sufficient data evidence that describe how more dependency on traditional medicine leads to poor access to biomedical maternity care.

 -Presenting a model suggesting these relations may contribute to enhance institutional racism, if the reader misunderstand that traditional medicine and cultural practices (and not racism directly) are the reason Garo women do not seek bio-medical maternal health care.

 -The solution would be to adjust the model, identifying the relations reported by Garo women only: 

 a) Poor access to biomedical maternal healthcare is rooted in institutional and interpersonal; b) there does not seem to be reports of internalized racism. If so, remove it from the Figure. c). the emergent theme (related to strong Cultural traditional-medicine practices / women agency) may work as a resistance towards both types of racism. 

4. Author may want to attenuate the affirmation of the hospital exploiting Garo women, taking into account a) ¿is it an experience of all the women in the study? (if so, author should report the abuse, if not, author could report the experience of some of the women with some of the staff) b) does the author want the hospital to be open to this study suggestions, and not defensive towards the claims?

 • P. 16: Adjust this sentence: “This illustrates a link to personally-mediated ethnocentrism, as presented in the following sub-section”. (Racism?) 

Reviewer's comments Discussion and Conclusion Section:

1. Results do not suggest how to solve the problem (implementing intercultural training programs) and it is not the scope of this kind of study. How ever, author can attenuate the affirmation, focusing on concluding facts that the study shows based on the evidence of the reports of Garo women: there is strong racism at different levels, and it strongly compromises biomedical healthcare access. Garo women count on a strong traditional medicine system to attend maternal healthcare. Garo women agency derives in their decision to avoid hospitals were they and their culture is mis-respected. Therefore, to increase biomedical healthcare access for Garo women racism needs to be prevent in the institutional and relational level. Author then can suggests the models to explore for future policy-making.

2. In discussion and abstract the author refers to Inclusive approaches like ‘Community Controlled Health Services’ and ‘Garo liaison officers’. These programs have not been previously presented, so the relation with result is unclear. Consider to present as examples, or expand it in the introduction. 

Response to the Reviewer's comments Discussion and Conclusion Section:

1. Dear respected reviewer,

I want to express my sincere gratitude for your insightful comments and recommendations. Your feedback has been instrumental in refining the manuscript, and I truly appreciate your time and expertise. I have incorporated your suggestions into the discussion section to provide a more nuanced and evidence-based conclusion.

In light of your input, I have adjusted the discussion as follows:

Based on the evidence from the reports of Garo women, it is evident that racism exists at different levels and significantly compromises their access to biomedical healthcare. Garo women have a strong reliance on traditional medicine for maternal healthcare. Their agency is rooted in their decision to avoid hospitals where they and their culture are not respected. Therefore, to enhance biomedical healthcare access for Garo women, it is imperative to address racism at both the institutional and relational levels. “then I suggest for implementing intercultural competency training program (CICTP) for the clinicians”.

2. Dear respected reviewer,

Thank you for your comment regarding the mention of 'Community Controlled Health Services' and 'Garo liaison officers' in the discussion and abstract sections of the manuscript. I appreciate your feedback and have made the necessary revisions based on your suggestion.

I have removed the policy recommendations related to these inclusive approaches from the discussion section as they were not adequately introduced or connected to the study's findings. Instead, I have focused on highlighting the broader implications of the study's results and the need for addressing racism at different levels within healthcare institutions, as per your earlier recommendations. This change was made to ensure clarity and relevance in the discussion section.

Once again, I want to express my gratitude for your valuable input, which has contributed to improving the coherence and effectiveness of the manuscript.

Other minor revisions as suggested by the reviewer: 

Here are some fragments that need writing revisions, but a final review of the writing is required. 

• Revise last phrase on p.4 “It is argues that a consideration of the women's varied social contexts and diverse identities is essential in comprehending their maternity care experiences (44).”

• Revise redaction of Jones’s ‘three levels of racism’ new paragraph in introduction.

• P. 5. Revise this redaction: “Given such strength, Jones’s theory, evident in studies like Janevic’s on Romani women(43) and others on Indigenous women (3,39,64), highlights racism's impact on maternal healthcare…” and “The Upazila hospital has medical staff, including non-Garo male gynecologist, with most of the pregnant patients seeking care there are not Garo women”.

• Positionality: Consider including this section with no title, but merged with the setting or in the suggested section of trustworthiness. In this phrase “The author has a distinct positionality concerning the research site and the participants” what do you mean by distinct positionality?

• p.9 This process enhanced trust and credibility between research assistants and participants during data collection, with the research assistants reporting to the author.

• P. 14. “As another participant shared, the issue of culturally inappropriate care extends to the midwifery meeting”. Do you mean “to the participation of the midwife ”?

• P14. This explanation may be removed as it is contains in previous paragraph: “This participant expresses concern about the absence of Garo female doctors, as seeking care from a non-Garo male doctor goes against their cultural norms of not sharing personal health problems with male doctors”.

• P.16. Differentiate researchers’ voice from participants: “Distance cost isn’t all. Hospitals want money. Despite their claim, they don’t charge 5 Taka (Bangladeshi currency). They demand more. Traveling makes you hungry, but food is expensive. We must spend all we earn if we go. She also adds, we don't control travel and treatment costs. They ignore our issue. They said ‘you have to follow the hospital rules’ whenever we told them our problem. What else can we do but avoid it? (L2, Middle-class, Age-30).

• P.21. Consider removing this part of the fragment that does not add much to what is being shown: “I don’t know. Probably would have sent me to a free hospital maybe” (L7, Low-class, Age-27).

Author's response:

• These studies often lack a context-specific understanding of intersectionality and overlook the perspectives of Indigenous women who face compounded prejudice and implicit bias due to various aspects of their identity such as gender and economic class. To comprehensively grasp Indigenous women’s maternity care experiences, it is imperative to consider their diverse identities within distinct social contexts. 

• The study employs Jones’s ‘three levels of racism' framework (22) along with Crenshaw’s intersectional lens (19) to frame and analyze the themes emerged from participants’ discussion. Jones’s model categorizes racism into three levels: institutional, personally-mediated, and internalized (22). This breakdown helps providing a deeper understanding of the complex impact of racism on maternal healthcare access. Unlike models that primarily focus on the social construction of race or ethnicity and the structural factors perpetuating racial inequalities (64–66), Jones’s comprehensive approach offers a holistic understanding. It suggests that institutional and personally-mediated racism contribute to the internalization of racism within the context of maternal healthcare access (22,28). Given its strong theoretical foundation, Jones’s model is particularly relevant for our study. 

• Studies such as Janevic’s research on Romani women (45) and similar work with Indigenous women (3,31,67,68), provide significant insights into the influence of racism on maternal healthcare access. However, these studies often neglect a crucial aspect: the intersection of racism with various identities, including gender and economic class. Despite its importance, the impact of this intersectionality on maternal healthcare access is frequently underemphasized (19,69). Maternal health research (70,71) underscores the significance of intersectionality, as it explores how social identities intersect to shape experiences, exacerbate healthcare disparities, and reveal interconnected systems of oppression (19). Hence, in alignment with Jones’s ‘Three Levels of Racism’ theory (22), the current study utilized intersectionality (19) as an analytical framework. It facilitates a clear understanding of how racism intersects with gender and economic class, leading to diverse experiences of discrimination that influence maternal healthcare access (19,69,72). This approach enhances the comprehension of healthcare inequalities and provides valuable insights for addressing the impact of racism on Garo Indigenous women’s access to maternal healthcare.

• I merge the ‘positionality section’ with setting.

• This strengthened trust and credibility between the research assistants and participants.

• Dear respected reviewer, I do not mean “to the participation of the midwife ”. I revised the sentences as follow to ensure clarity: Another participant also mentioned that culturally inappropriate care includes not being able to meet the birth attendant before delivering in a hospital, which goes against their cultural norms and expectations. 

• Dear respected reviewer, I removed this sentence: “This participant expresses concern about the absence of Garo female doctors, as seeking care from a non-Garo male doctor goes against their cultural norms of not sharing personal health problems with male doctors”.

• Dear respected reviewer, in the revised manuscript I tried by best to differentiate author’s voice from the participants. 

• Dear respected reviewer, I removed this fragmented quote: “I don’t know. Probably would have sent me to a free hospital maybe” (L7, Low-class, Age-27). 

I trust that these alterations have enhanced the overall quality and coherence of the manuscript, aligning it more closely with the standards of the journal.

Once again, thank you for your time, expertise, and commitment to improving the research. Your feedback has been instrumental in strengthening this work.

Kind Regards.

The Author

---

## [Decision Letter · Decision Letter 3]

27 Sep 2023

PONE-D-22-34953R3Racism and Access to Maternal Health care Among Garo Indigenous Women in Bangladesh: A Qualitative Descriptive StudyPLOS ONE

Dear Dr. Chowdhury,

Thank you for submitting your manuscript to PLOS ONE. After careful consideration, we feel that it has merit but does not fully meet PLOS ONE’s publication criteria as it currently stands. Therefore, we invite you to submit a revised version of the manuscript that addresses the points raised during the review process.

Thank you for addressing the comments from the peer review. However, there are still minor revisions needed. Please review the reviewer's comment below and the attached PDF for additional feedback. One important matter regards the use of the term "exploitation". It's unclear and doesn't correspond to the additional costs faced by Garo women due to racism, as stated in the manuscript. 

We look forward to receiving your revised manuscript.

Kind regards,

Ivan Sarmiento

Academic Editor

PLOS ONE

Journal Requirements:

Reviewers' comments:

Reviewer's Responses to Questions

**Comments to the Author**

1. If the authors have adequately addressed your comments raised in a previous round of review and you feel that this manuscript is now acceptable for publication, you may indicate that here to bypass the “Comments to the Author” section, enter your conflict of interest statement in the “Confidential to Editor” section, and submit your "Accept" recommendation.

Reviewer #3: All comments have been addressed

2. Is the manuscript technically sound, and do the data support the conclusions?

Reviewer #3: Yes

3. Has the statistical analysis been performed appropriately and rigorously? 

Reviewer #3: N/A

4. Have the authors made all data underlying the findings in their manuscript fully available?

Reviewer #3: Yes

5. Is the manuscript presented in an intelligible fashion and written in standard English?

Reviewer #3: Yes

6. Review Comments to the Author

Reviewer #3: The author has achieved a high quality report for a study of great relevance to Garo women, which also contributes to health equity work for indigenous women in general.

Minor corrections are mainly focused on the final revision of the English language writing.

1. Pay attention in the methodological section that the verb ending is uniform in the past tense.

2. On page 18, the methodological design is qualitative descriptive approach. No qualitative description.

3. Trustworthiness (p.10) Author can redact this section, in a simpler way: The strategies to enhance trustworthiness in this study where…

4. Ethical considerations (P.22). Replace “privacy” with “anonymity”.

Results:

5.(P. 27). Correct the capital “P” in Participants.

6.The last quote of the participants (p.23) is significantly relevant: not only it shows women agency, but also indicates the perceived efficacy of the traditional medicine in women healthcare.

7.(p. 25) Please briefly clarify the discussion with the Ethiopia study and the other study about identity based racial discrimination at the institutional level. It still not clear how does this study contrast with the study in Ethiopia.

7. PLOS authors have the option to publish the peer review history of their article (what does this mean?). If published, this will include your full peer review and any attached files.

Reviewer #3: No

---

## [Author Response · Author response to Decision Letter 3]

21 Oct 2023

Dear Esteemed Reviewer,

I wish to convey my heartfelt appreciation for your invaluable feedback regarding my manuscript. Your insights and recommendations have played a pivotal role in enhancing the quality and clarity of the research. I am delighted to inform you that I have diligently addressed each of your suggestions, and the changes have been seamlessly incorporated into the revised manuscript (please see the lines in red color in the manuscript). Additionally, I have also addressed the revisions you suggested in the manuscript (please see the revision highlighted in green color in the manuscript). To offer a clear and organized overview of the modifications, I have compiled them as below:

1. Reviewer: Pay attention in the methodological section that the verb ending is uniform in the past tense.

Author: Dear respected Reviewer,

I want to express my gratitude for your thorough review and attention to detail. Ensuring consistency in the verb endings in the past tense throughout the methodological section is an essential aspect of maintaining the quality of the paper. I carefully reviewed and revised the section to address this concern, and I appreciate your valuable input in maintaining the linguistic consistency and precision of the content.

2. Reviewer: On page 18, the methodological design is qualitative descriptive approach. No qualitative description.

Author: Dear respected Reviewer,

Thank you for your meticulous review of the manuscript. I appreciate your keen eye for detail. I will make the necessary correction on page 18, ensuring that the methodological design is referred to as a "qualitative descriptive approach" rather than "qualitative description." Your feedback is essential in maintaining precision and clarity in the content, and I'm grateful for your contribution to the refinement of the paper.

3. Reviewer: • Trustworthiness (p.10) Author can redact this section, in a simpler way: The strategies to enhance trustworthiness in this study where…

Author: Dear respected Reviewer,

I want to express my gratitude for your thoughtful review and suggestions. I agree that the section on trustworthiness (on page 10) can be revised to enhance clarity and simplicity. I made the necessary revisions to present the strategies for ensuring trustworthiness in a more straightforward and reader-friendly manner. Your feedback is highly valuable, and I appreciate your contribution to improving the manuscript. [please see the revised section ‘trustworthiness’. 

4. Reviewer: • Ethical considerations (P.22). Replace “privacy” with “anonymity”.

Author: Dear respected Reviewer,

I want to express my gratitude for your careful review of the manuscript. Your suggestion to replace "privacy" with "anonymity" in the section on ethical considerations (on page 22) is well-noted and appreciated. I agree that the term "anonymity" is more accurate and aligns better with the ethical considerations discussed in that section. I made the recommended change to enhance the precision of the content and ensure it accurately reflects the research ethics.

Thank you for your valuable input, which contributes to the overall quality of the paper.

5. Reviewer: 

Results:

 • (P. 27). Correct the capital “P” in Participants.

Author: • Dear respected Reviewer,

Thank you for your careful review and for bringing to my attention the capitalization issue with "Participants." I apologize for the oversight, and I made the necessary correction to ensure consistency in capitalization throughout the document. Your attention to detail is greatly appreciated, and it helped improve the overall quality of the manuscript.

 • The last quote of the participants (p.23) is significantly relevant: not only it shows women agency, but also indicates the perceived efficacy of the traditional medicine in women healthcare.

Author: • Dear respected Reviewer,

Thank you for your kind and valuable observation. I appreciate your observation and recognition of the significance of the last quote from the participants on page 23.

 • (p. 25) Please briefly clarify the discussion with the Ethiopia study and the other study about identity based racial discrimination at the institutional level. It still not clear how does this study contrast with the study in Ethiopia

Author: • Dear respected Reviewer,

Thank you for your valuable comment. I appreciate your interest in further clarification regarding the contrast between this study and the research in Ethiopia. To provide a more convincing explanation, it’s important to emphasize that in this study conducted in Bangladesh, the primary obstacle for Garo Indigenous women’s maternal healthcare access is institutional racism within healthcare institutions, which stems from Indigenous identity-based discrimination. In contrast, in Ethiopia, the challenges faced by Indigenous women were more closely related to the actual unavailability of healthcare resources, along with cultural norms and a shortage of female healthcare professionals. By underlining these differences, I aim to illustrate how various factors contribute to racial discrimination in distinct regional contexts. [please see page 25, ‘discussion section’, lines in red color] 

Thank you for your understanding and guidance. 

Your diligent and thoughtful review has been instrumental in improving the manuscript, and I am deeply grateful for your contribution.

Sincerely,

The Author

---

## [Editor Report · Decision Letter 4]

25 Oct 2023

PONE-D-22-34953R4Racism and Access to Maternal Health care Among Garo Indigenous Women in Bangladesh: A Qualitative Descriptive StudyPLOS ONE

Dear Dr. Chowdhury,

Thank you for submitting your manuscript to PLOS ONE. After careful consideration, we feel that it has merit but does not fully meet PLOS ONE’s publication criteria as it currently stands. Therefore, we invite you to submit a revised version of the manuscript that addresses the points raised during the review process.

We look forward to receiving your revised manuscript.

Kind regards,

Ivan Sarmiento

Academic Editor

PLOS ONE

Journal Requirements:

**Additional Editor Comments:**

Thank you for addressing the comments from the reviewer. However, you have not addressed my comments. You will find them in the attached PDF. You can also read the email with my previous decision for additional details. In your response letter, I enourage you to use a direct style to facilitate the revision. You need to pay particular attention to the concept of explotation that you use because it is not evident how it corresponds to the economic dificulties that you describe in the reslts section. There is an important difference between having extra costs and being subject to explotation.

---

## [Author Response · Author response to Decision Letter 4]

4 Nov 2023

Dear Respected Academic editor, I have attached a separate file compiling all of my responses to your's and the reviewer's comments.

---

## [Editor Report · Decision Letter 5]

8 Nov 2023

Racism and Access to Maternal Health care Among Garo Indigenous Women in Bangladesh: A Qualitative Descriptive Study

PONE-D-22-34953R5

Dear Dr. Chowdhury,

We’re pleased to inform you that your manuscript has been judged scientifically suitable for publication and will be formally accepted for publication once it meets all outstanding technical requirements.

Kind regards,

Ivan Sarmiento

Academic Editor

PLOS ONE
---

## [Editor Report · Acceptance letter]

17 Nov 2023

PONE-D-22-34953R5 

Racism and Access to Maternal Health care Among Garo Indigenous Women in Bangladesh: A Qualitative Descriptive Study 

Dear Dr. Chowdhury:

I'm pleased to inform you that your manuscript has been deemed suitable for publication in PLOS ONE. Congratulations! Your manuscript is now with our production department. 

Kind regards, 

on behalf of

Dr. Ivan Sarmiento 

Academic Editor

PLOS ONE